# Heat-Killed Lactobacilli Preparations Promote Healing in the Experimental Cutaneous Wounds

**DOI:** 10.3390/cells10113264

**Published:** 2021-11-22

**Authors:** Wan-Hua Tsai, Chia-Hsuan Chou, Tsuei-Yin Huang, Hui-Ling Wang, Peng-Ju Chien, Wen-Wei Chang, Hsueh-Te Lee

**Affiliations:** 1Research and Development Department, GenMont Biotech Incorporation, Tainan 741014, Taiwan; twh@genmont.com.tw (W.-H.T.); chc@genmont.com.tw (C.-H.C.); patty@genmont.com.tw (T.-Y.H.); 2School of Biomedical Sciences, Chung Shan Medical University, Taichung 402306, Taiwan; huilin117@gmail.com (H.-L.W.); chienpengju@gmail.com (P.-J.C.); 3Department of Medical Research, Chung Shan Medical University Hospital, Taichung 402306, Taiwan; 4Institute of Anatomy & Cell Biology, National Yang Ming Chiao Tung University, Taipei 112304, Taiwan; 5Taiwan International Graduate Program in Molecular Medicine, National Yang Ming Chiao Tung University and Academia Sinica, Taipei 115024, Taiwan; 6Brain Research Center, National Yang Ming Chiao Tung University, Taipei 112304, Taiwan

**Keywords:** heat-killed probiotics, Lactobacillus, *L. plantarum*, *L. paracasei*, skin wound healing, lipoteichoic acid

## Abstract

Probiotics are defined as microorganisms with beneficial health effects when consumed by humans, being applied mainly to improve allergic or intestinal diseases. Due to the increasing resistance of pathogens to antibiotics, the abuse of antibiotics becomes inefficient in the skin and in systemic infections, and probiotics may also provide the protective effect for repairing the healing of infected cutaneous wounds. Here we selected two Lactobacillus strains, *L. plantarum* GMNL-6 and *L. paracasei* GMNL-653, in heat-killed format to examine the beneficial effect in skin wound repair through the selection by promoting collagen synthesis in Hs68 fibroblast cells. The coverage of gels containing heat-killed GMNL-6 or GMNL-653 on the mouse tail with experimental wounds displayed healing promoting effects with promoting of metalloproteinase-1 expression at the early phase and reduced excessive fibrosis accumulation and deposition in the later tail-skin recovery stage. More importantly, lipoteichoic acid, the major component of Lactobacillus cell wall, from GMNL-6/GMNL-653 could achieve the anti-fibrogenic benefit similar to the heat-killed bacteria cells in the TGF-β stimulated Hs68 fibroblast cell model. Our study offers a new therapeutic potential of the heat-killed format of Lactobacillus as an alternative approach to treating skin healing disorders.

## 1. Introduction

Studies that focus on the biological functional integration between microbiota and the human body have implied that the microbiome may have a principal impact on multiple physiological connections, including protection against injury-induced infections and modulation in the body. The importance of commensal microorganisms in maintaining host health has been recognized early and focused on intestinal microbiota research. The application of a wider and deeper potential to the microbiota is a burning issue for healthcare. As with the role of local and systemic probiotic effects, approaches that manipulate its composition to improve host disorders have become increasingly important. An earlier study indicated that probiotics altered the healing capacity of mucous wounds [1] and were reported to have a potentially beneficial effect on inflammatory cutaneous conditions [2]. Moreover, probiotics are an alternative medication and are often considered to promote health care [3]. *Lactobacillus* is a common probiotic species, and its health benefits for human disease improvement have been widely reported [4,5,6].

Wound healing replaces damaged or damaged tissue with tissue newly generated by a living organism [7]. The healing process involves four stages: hemostasis, inflammation, proliferation, and tissue remodeling [8,9]. Interfering with the wound healing process can cause chronic wounds, which do not heal as expected, or remain in the healing process for more than 4–6 weeks [10,11]. Most chronic wounds exhibit ulcers and are associated with diseases, such as ischemia and diabetes mellitus; the number of patients with chronic wounds in the United States is approximately 3–6 million each year [8]. Consequently, the development of materials that promote wound healing has medical and business value.

Previous research has shown that excessive infiltration of neutrophils and macrophages into wound tissue sites often delays healing [12,13], and abnormal expression of proinflammatory factors, such as interleukin-1β (IL-1β) and tumor necrosis factor-α (TNF-α) can also be observed at wound sites [9,14,15]. The overexpression of these proinflammatory factors increases matrix metalloproteinases and then decomposes extracellular matrix (ECM), such as collagen, which affects the tissue remodeling phase and causes chronic wounds in the wound tissue site [16,17]. In addition, clinical human specimens of chronic venous ulcers found that transforming growth factor-β (TGF-β) and its downstream signaling molecule, phosphorylation of Smad2 were inhibited [18]. In the acute early stage of wound healing, keratinocytes enter wound sites to cause re-epithelialization; TGF-β may inhibit the proliferation of keratinocytes and promote wound healing [19,20]. Furthermore, administering TGF-β to human skin fibroblasts can induce collagen synthesis [21].

The topical use of probiotics in the improvement of skin wound repair has been reported by several studies [22]. Most of the studies used live probiotics but it may become opportunistic infection when used in at-risk population groups, such as people with weakened immunity [23] and the heat-killed format could avoid such risk. In this study, we reported that the heat-killed *Lactiplantibacillus plantarum* strain of GMNL-6 (called GMNL-6) or *Lacticaseibacillus paracasei* strain of GMNL-653 (called GMNL-653) containing gel could reduce the injury area of the experimental tail epidermis wound in mice. In addition, the key component within the heat-killed GMNL-6 or GMNL-653 probiotics preparations were also demonstrated that was associated with the cell wall-derived lipoteichoic acid (LTA) through decreasing the TGF-β/Smad induced α-smooth muscle actin (α-SMA) expression in Hs68 human fibroblasts.

## 2. Materials and Methods

### 2.1. Reagents and Probiotics Preparations

For this study, *Lact**iplantibacillus plantarum* GMNL-6 (BCRC910777) and *Lact**icaseibacillus paracasei* GMNL-653 (BCRC910721) and other bacteria (Appendix A) were obtained from GenMont Biotech Incorporation, Inc., Taiwan. The bacteria were extracted and purified from the human intestine, isolated, and maintained under facultative anaerobic conditions. The heat-killed preparation of GMNL-6 or GMNL-653 was made by washing grown bacterial cells with distilled H_2_O and suspended as 10^10^ cells/mL in distilled H_2_O followed by autoclave (121 °C for 30 min). Recombinant human transforming growth factor-β (TGF-β) (MilliporeSigma, Burlington, MA, USA) was used as a central mediator of active fibrogenesis. All reagents and solvents were used for research purposes.

### 2.2. Tail Full-Thickness Wounding Surgery in a Mouse Model

This animal study conformed to the Guide for the Care and Use of Laboratory Animals (Institute of Laboratory Animal Resources, eighth edition, 2011), and all animal experiments were approved by the Animal Care and Utilization Committee of GenMont Biotech Incorporation, Inc. (Taiwan IACUC Approval No: 194; trial no: 105006). A study was conducted for 20 days. We used 8-week old female BALB/c mice (*n* = 4 for each group) who were anesthetized with isoflurane (3% for induction, 1% for maintenance) during surgery, and the wounds of 0.2 cm × 0.1 cm were introduced from 1 cm from the root of mice tails. The newly introduced wounds waited for one day to stop the aseptic gauze followed by covering gels (the gel used in this study was Ulora Gel (BEE BRAND MEDICO DENTAL CO., LTD., Osaka, Japan). The composition of Ulora Gel for 1 g was listed as followed: xanthangum 0.02 g, xylitol 0.06 g, trehalose 0.04 g, glycerol 0.1 g, sterilized H_2_O 0.28 g. with or without heat-killed GMNL-6 or GMNL-653 bacterial cells at a final concentration of 1 × 10^10^ cells/g of gel. For the experimental procedure, the tail wounds were smeared with 0.01 g gel evenly per day (i.e., 1 × 10^8^ bacterial cells/time) for a total of five days (from day-1 to day-5 after tail skin surgery). To make sure they completely absorbed the gels, they were left standing for 15–20 min after spreading the gel. The tails of mice were imaged and recorded at 1, 3, 5, 10, 15, and 20 days after wounding. At the end of the experiment, mice were euthanized with CO_2_. The mouse tails were isolated and subjected to histological analysis or stored at −80 °C.

### 2.3. Cell Culture

Human foreskin fibroblasts (Hs68, CRL-1635) were obtained from American Type Culture Collection (ATCC, Manassas, VA, USA) and cultured in Dulbecco’s modified Eagle medium supplemented with 10% fetal bovine serum, penicillin (100 U/mL), and streptomycin (100 μg/mL) under steady-state conditions at 37 °C and 5% CO_2_ in a humidified incubator.

### 2.4. Synthesis Assay of Collagen

The Hs68 cells were seeded evenly in 6-well-plates (2 × 10^5^ cells/well), washed twice with phosphate-buffered saline (PBS), replaced with serum-free medium, and treated with various heat-killed Lactobacillus strains at 10-fold (1 × 10^9^ cells/well) and 20-fold (5 × 10^8^ cells/well) dilutions for 24 h. The supernatant was collected for analysis using a Human Collagen I alpha 1 ELISA Kit (Novus Biologicals LCC, Centennial, CO, USA).

### 2.5. Detection of the Neuronal Amine Synthase Ability

The Hs68 cells were seeded evenly in 6-well-plates (2 × 10^5^ cells/well) and treated with various heat-killed Lactobacillus strains at 10-fold (1 × 10^9^ cells/well) and 20-fold (5 × 10^8^ cells/well) dilutions for 24 h. The cell RNA was extracted and converted into cDNA, and the serine palmitoyltransferase small subunit A (SPTSSA) and β-actin were quantified and normalized using real-time PCR analysis. The sequences of primers (5’ to 3’) were listed below:

SPTSSA

Forward: TGGTTCTACTACCAGTACCTGC

Reverse: CTGGGGCATGAAGACGTATC

β-actin

Forward: CCTTGGCATCCACGAAACT

Reverse: TCTCCTTCTGCATCCTGTCG

### 2.6. Western Blot Analysis

Aliquots (50 µg) of cell lysates were resolved using 10% acrylamide gels of sodium dodecyl sulfate–polyacrylamide gel electrophoresis (SDS-PAGE) and then transblotted onto an Immobilon^TM^-P membrane (Millipore, Bedford, MA, USA). After blocking with 5% skimmed milk, the membranes were incubated with various primary antibodies, and immunoreactivity was detected using horseradish-conjugated secondary antibody and visualized using an enhanced chemiluminescence kit (PerkinElmer, Boston, MA, USA). The following primary antibodies were used: mouse monoclonal anti-αSMA antibody (1:1000, Cat. No. sc-32251, Santa Cruz Biotechnology, Inc., Dallas, TX, USA), rabbit monoclonal anti-Smad 2/3 (1:1000, Cat. No. 04-914, EMD Millipore Corporation, Temecula, CA, USA), rabbit monoclonal anti-Phospho-Smad2 antibody (1:1000, Cat. No. 04-935, EMD Millipore Corporation), mouse monoclonal anti-human MMP1 antibody (1:1000, Cat. No. MAB3307, EMD Millipore Corporation), mouse monoclonal anti-MMP2 antibody (1:1000, Cat. No. MAB3308, EMD Millipore Corporation), and rabbit polyclonal anti-GAPDH antibody (1:1000, GeneTex International Corporation, Hsinchu City, Taiwan).

### 2.7. Histological and Immunohistochemical Assessment

Tail tissue blocks were cut into 5-μm sections. For general histological examination, deparaffinized sections were stained with hematoxylin and eosin (H&E). The distance of the distance between the epithelial migration tongues (dMT) or the distance between the hair follicles (dA) was determined as the descriptions from Gerharz et al. [24] using ImageJ software (Version 1.53 m, National Institutes of Health, Bethesda, MA, USA). The ratio of re-epithelialization was calculated by the formula of dMT/dA. For immunohistochemical staining, the deparaffinized tail sections were reacted with 3% H_2_O_2_ for 20 min. After blocking with 2% bovine serum albumin for 1 h at 25 °C, samples were incubated with mouse monoclonal anti-human MMP1 antibody (1:100, Cat. No. MAB3307, EMD Milli-pore Corporation) and mouse monoclonal anti-αSMA antibody (1:100, Cat. No. sc-32251, Santa Cruz Biotechnology, Inc.) overnight at 4 °C and then with the corresponding secondary antibody for 1 h at 37 °C. Antigenic sites were visualized by administering 3,3′-diaminobenzidine and hematoxylin for counterstaining. All slides were scanned through a TissueFAXS PLUS System (TissueGnostics, Taborstrasse, Vienna, Austria).

### 2.8. Masson’s Trichrome Staining

Masson’s trichrome staining protocol was performed following the manufacturer’s standard procedure. After de-paraffining, the sections were immersed in Bouin’s solution for 2 h at 56 °C. After washing with PBS, samples were then stained with Wiegert’s iron hematoxylin for 15 min, incubated with Orange G for 1 min and with Masson’s B solution for 30 s, and washed with 1% acetic acid solution for at least 10 min. Finally, all sections were reacted with a phosphomolybdic-phosphotungstic acid solution, stained with aniline blue for 15 min, and then mounted with a mounting solution for further analysis. Images were obtained after scanning by a TissueFAXS PLUS System.

### 2.9. Statistical Analyses

Statistical analyses were performed using GraphPad Prism Software (version 7, San Diego, CA, USA). Data are presented as mean ± SEM. The unpaired *t*-test (for the comparisons between two groups) and two-way analysis of variance (ANOVA) (for the comparisons among groups with three or above) followed by Tukey’s multiple comparison test were used to evaluate statistical significance.

## 3. Results

### 3.1. Heat-Killed Lactobacillus GMNL-6 or GMNL-653 Provides Beneficial Effects in Skin Repairing

Both collagen and ceramide play a critical role during skin repair or connective tissue remodeling after injury [25]. Serine palmitoyltransferase activity is the rate-limiting step in ceramide biosynthesis [25,26]. Thus, we first evaluated the skin beneficial potential of heat-killed probiotics preparations of GMNL-6 or GMNL-653, by the determination of collagen protein synthesis, and the mRNA expression of serine palmitoyltransferase small subunit A (SPTSSA) in human foreskin fibroblasts, Hs68 cells. The results revealed that the collagen synthesis ability increased significantly after treatment of heat-killed GMNL-6 (Figure 1A, approximately 9-fold) or GMNL-653 (Figure 1B, approximately 7-fold). Similarly, the heat-killed GMNL-6 or GMNL-653 could also upregulate the gene expression of SPTSSA compared with vehicle treatment (Figure 1C for GMNL-6 and Figure 1D for GMNL-653). These data suggest that these two heat-killed probiotic preparations of GMNL-6 or GMNL-653 might have beneficial effects on skin repair.

### 3.2. TGF-β/Smad Signaling Involved in the Heat-Killed Lactobacillus GMNL-6 or GMNL-653 Promoted Wound Healing Ability

The major significant translational importance in the wound healing process includes the effect of TGF-β on fibroblast and keratinocyte migration, extracellular matrix (ECM) production and remodeling phase, and cross-talk between fibroblasts and keratinocytes. A better understanding of the wound healing process and the critical roles of different cells and signaling, such as TGF-β-related signaling, can lead scientists to develop more sophisticated therapies for wound repair and prevention of excessive scar formation, which is considered to be related to excess myofibroblasts within wound sites [27]. Based on this knowledge, we investigated the effect of heat-killed probiotics GMNL-6 or GMNL-653 preparation to TGF-β induced myofibroblast differentiation in Hs68 cells with the determination of αSMA expression. As shown in Figure 2, we found that eventually, the heat-killed probiotics GMNL-6 or GMNL-653 treatment at 5 × 10^8^ or 1 × 10^9^ cells/mL had an obviously inhibitory effect on the expression of αSMA, Smad2/3, and phosphorylated Smad2 in a dose-dependent manner. On the contrary, the treatment of heat-killed GMNL-6 or GMNL-653 preparation upregulated the synthesis of MMP1, which is the key MMP for re-epithelialization during skin wound healing [28] without interfering with MMP2 expression. These data imply that the heat-killed probiotics GMNL-6/GMNL-653 preparation display the beneficial potential to skin wound healing by promoting re-epithelialization and suppressing the excess accumulation of myofibroblasts.

### 3.3. Heat-Killed Probiotics Preparations of GMNL-6 or GMNL-653 Exhibit Excellent Wound Healing Ability in the Mouse Model of Experimental Tail Wounds

Due to the in vitro increasing of collagen synthesis and SPTSSA mRNA expression (Figure 1), and the MMP1 (Figure 2) in Hs68 cells, we further tested the in vivo skin repair efficacy of the heat-killed probiotic preparations of GMNL-6 or GMNL-653 by an experimental tail wound in mice according to a previous publication [29]. Briefly, the experimental design and time scheme used are illustrated in Figure 3A (see Section 2.2 for the detailed procedure descriptions). The tail full-thickness wound sites of the mice were excisional wounds. After the surgery, each mouse tail was observed and recorded using a digital camera from day-1 to day-20 after surgery (Figure 3B), the precise extent of the injury was measured with a ruler, and the statistical data of the wound after treatment of heat-killed GMNL-6 or GMNL-653 preparation were compared with the vehicle group. Interestingly, both preparations of heat-killed GMNL-6 and GMNL-653 significantly accelerated the wound healing on the tail wounds in comparison to those covered with vehicle gel (Figure 3C). However, GMNL-653 gel displayed a better activity in skin wound repair than GMNL-6. The significantly accelerated wound closure at day-3 and day-15 was only observed in the GMNL-653 group, but both GMNL-6 and GMNL-653 gels recovered the tail wounds at day-20, whereas the wound area remained more than 20% in the vehicle group (Figure 3C). Altogether, these findings suggest that the heat-killed Lactobacillus GMNL-6 or GMNL-653 exhibited excellent wound healing ability on the experimental tail wound in mice after five days of treatment.

### 3.4. Heat-Killed Probiotics GMNL-6 or GMNL-653 Preparations Provide Prominent Tail-Wound Healing Recovery Ability

Collective cell migration is an important step in essential physiological healing processes, and injured site-associated cells often move as a tightly or loosely associated cohesive group. We next evaluated the effect of heat-killed probiotics GMNL-6 or GMNL-653 gels on the distance of migrating epithelial tongues (dMT) or the distance between hair follicles (dA) respectively, according to the report from Gerharz et al. [24] (Figure 4A–C). The re-epithelialization ratio of wounds was calculated by dMT/dA (Figure 4D). The quantification results revealed that gels containing the heat-killed probiotics GMNL-6 or GMNL-653 could effectively reduce the dMT measurement from day-5 to day-9 (Figure 4B). Similar results were observed for wound contraction, which was evaluated by dA measurements (Figure 4C). However, the ratio of re-epithelialization was increased in the GMNL-6 group on day-5 and day-9 (Figure 4D). These findings suggest that treatment with the heat-killed probiotics GMNL-6 or GMNL-653 preparations provides an evident tail-wound healing recovery ability.

### 3.5. Heat-Killed Probiotics GMNL-6 or GMNL-653 Gel Preparations Increase MMP-1 Expression in the Early Stage (Day-5) and Decrease the αSMA Expression in the following Stage (Day-9) in the Tail-Wound Healing Mouse Model

The tissue remodeling stage is the final phase of wound healing after the initial injury. In skin wound healing, MMP-1 is required for keratinocyte migration along with type I collagen during the re-epithelialization stage [15,16,17,18]. Therefore, we further evaluated MMP1 expression in the early stage (day-5) and in the following stage (day-9) after experimental wound introduction. Our data showed that heat-killed probiotics gel preparations of GMNL-6 or GMNL-653 increased MMP-1 expression in the early stage (day-5) but gradually decreased in the following stage (day-9) (Figure 5A). More importantly, the structured collagen fibers were observed in the dermis of heat-killed probiotics GMNL-6, or GMNL-653 gel treated wounds accompanying the decreased expression of αSMA, a marker for myofibroblasts, at a later stage of day 9 (Figure 5B) indicating that the promotion of skin wound healing without excessive scar formation of these two heat-killed preparations. These data indicate that the heat-killed probiotics GMNL-6 or GMNL-653 gel preparations not only provide apparent tail-wound healing recovery ability but also prevent excessive skin fibrosis.

### 3.6. The Lipoteichoic Acid (LTA) from Cell Wall of GMNL-6 or GMNL-653 Cause the Similar Beneficial Effects to the TGF-β Induced Fibrosis of Hs68 Cells

We next tried to identify the functional component of heat-killed probiotics GMNL-6 or GMNL-653 preparation in promoting skin wound healing. Lipoteichoic acid (LTA) and peptidoglycan (PGN) are the major components of the cell wall of Gram-Positive bacteria, which are pivotal components for beneficial effects on their host [30,31]. Therefore, we further purified LTA or PGN from GMNL-6 or GMNL-653 and used them to treat Hs68 cells in combination with TGF-β stimulation. Our data showed that treatment with the GMNL-6 or GMNL-653 derived LTA, but not PGN, suppressed the expression of p-Smad2^ser465/467^ and αSMA at a similar level to those of using heat-killed GMNL-6 or GMNL-653 bacterial cells in TGF-β stimulated Hs68 cells (Figure 6A,B). These data demonstrate that the beneficial effects of heat-killed probiotics GMNL-6 or GMNL-653 preparations in skin wound healing could be mediated through the cell wall component of LTA.

## 4. Discussion

In this study, we demonstrated that heat-killed probiotics GMNL-6 and GMNL-653 preparations help to heal skin injuries in the in vitro model of TGF-β stimulated Hs68 human fibroblast cells and in the in vivo model of experimental tail wounds in mice. Wound healing refers to the process of replacing destroyed or damaged tissue with newly produced tissue [7]. The wound-healing process can be broadly distinguished into four stages: (1) TGF-β upregulation-induced inflammatory effects in the wound area, (2) fibroblast migration, (3) extracellular matrix (ECM) production, accumulation in the remodeling phase, and finally (4) cross-talk between fibroblasts and other cell types in injury neighborhood areas. Therefore, a deeper understanding of the wound healing process and the critical roles of different cells and molecules will allow scientists to find more sophisticated therapies for wound repair and preventing excessive fibrosis formation.

Compared to the report from Falange et al. [32], the increased infiltrating inflammatory cells and slower healing rate of tail wounds were found in the vehicle group of our study (Figure 3 and Figure 4). The differences may result from two factors. First, we did not apply antibiotics in wound management, but the study of Falange et al. used the antibiotics trimethoprim sulfa [32]. As a systemic review of the effects of antibiotics in animal skin wounds, Altoé et al. suggested that antibiotics generally reduced wound healing time which was associated with reducing inflammatory cells and increasing the proliferation of fibroblasts [33]. Second, the mouse strains may cause a difference in wound healing rate. The mouse strain used in the study of Falange et al. was 129SV × C57BL/6 [32], whereas we used BALB/c mice. From the report of Li et al., the wound healing rate of ear holes in 129J or C57BL/6 strain of mice was much higher than BALB/c [34]. Similar observations were found in the healing of corneal wounds by the report from Pal-Ghosh et al., in which they found that the migration capability of corneal epithelial cells from BALB/c mice was 60% lower than those from C57BL/6 [35]. Nevertheless, under the same wound management condition, heat-killed GMNL-6 or GMNL-653 gels showed good improvement.

Probiotics play critical roles in maintaining health and balance among different communication organs and tissue repair. Therefore, we would like to explore a more potential therapeutic approach for probiotics related to healthy skin. Few studies have reported that functional and probiotic components may improve collagen synthesis and increase skin hydration [25,36]. Thus, we first screened several probiotic strains of *Lactobacillus* spp. according to the promotion of collagen synthesis in Hs68 cells (Appendix A) and finally selected the GMNL-6 and GMNL-653 to further investigate their potential in wound healing. In the present study, the heat-killed probiotics GMNL-6 or GMNL-653 preparations could provide the benefits on skin wound healing with the possible mechanisms including promoting collagen synthesis to moisturize in the earlier healing stage (Figure 1 and Figure 2) and increasing MMP1 to accelerate re-epithelialization in the wound site at the earlier healing stage in our animal study (Figure 2 and Figure 5).

Increased collagen synthesis and hydration are important issues in the healing study of injuries but finding the best time to administer medications is often confusing for scientists. Too early administration can result in excessive fibrosis, such as scarring, whereas late administration can reduce the therapeutic effect. Here, our evidence demonstrated that applying heat-killed probiotics GMNL-6 or GMNL-653 gel preparations at an earlier stage (day-1 to day-5) might accelerate wound healing (Figure 2B,C) and increase wound re-epithelialization (Figure 4). Furthermore, we found that heat-killed probiotics GMNL-6 and GMNL-653 treatment-induced MMP-1 expression in the early stage (day-5), downregulation in the later stage (day-9), and additional evidence showed that αSMA had lower expression, but connective tissue was still located in the wound sites on day-9. αSMA is a marker for myofibroblasts and the physiological function of these cells is to cause wound contraction thereby reducing wound margin [37]. Myofibroblasts should be removed through the induction of apoptosis during the wound healing process to prevent excess wound contraction, scarring, and fibrosis [37]. However, the skin wound healing in Acta2 knockout mice was as good as the wild-type mice, which illustrates that myofibroblasts are not absolutely necessary for wound contraction [34,38]. It is also known that there is a heterogeneity of dermal fibroblasts that affects normal or pathological skin wound healing [39]. The observations of the reduction of αSMA expression in mouse tail wounds at Day-9 (Figure 5) but accelerated wound contraction (Figure 4) by heat-killed GMNL-6 or GMNL-653, suggests that heat-killed GMNL-6 or GMNL-653 may have different effects on different subpopulations of dermal fibroblasts, but it remains to be further investigated.

*Staphylococcus aureus* accounts for the most common bacterial isolate among human skin wounds [40], and the formation of *S. aureus* biofilm infection on dorsal skin wounds in mice has been reported to delay the wound repairing [41]. We have previously demonstrated that the heat-killed GMNL-6 displayed an inhibitory in the formation of *S. aureus* biofilm [25]. Here we also found that the heat-killed GMNL-653 could cause aggregation with *S. aureus* that formed precipitates in a test tube when mixing them together (Appendix A) and the aggregation between heat-killed GNML-653 and *S. aureus* was further confirmed by scanning electron microscope (Appendix A). We also observed that heat-killed GMNL-653 suppressed the adhesion of *S. aureus* on Hs68 cells (Appendix A). The treatment of GMNL-653 derived LTA reduced the formation of *S. aureus* biofilm (Appendix A). From these observations, the beneficial effects of heat-killed GMNL-6 or GMNL-653 on skin wound repair may also include their inhibitory effect on *S. aureus* biofilm formation through the cell wall derived LTA.

According to our previous study [25], the major cell wall components of heat-killed probiotics, such as LTA, can provide skincare benefits. In this study, similar findings are also demonstrated in Figure 6. In addition, we demonstrated that the beneficial effect of heat-killed probiotic preparations could achieve by suppressing TGF-β/Smad2 signaling (Figure 6A,B). The heat-killed probiotics preparations can avoid the risk of infection in immunocompromised subjects, such as the medications of glucocorticoid, which is widely used in inflammatory diseases but is known to inhibit wound repair [42]. Chronic wounds with slowly healing are often observed in diabetes patients [43], whose immune system has weakened with the increased incidence of infectious diseases [44]. Due to the non-infectious characteristics of heat-killed GMNL-6 or GMNL-653 preparations, their potential in chronic skin wound care is worthy of further investigation in the future.

## 5. Conclusions

In conclusion, the results of our study are as follows: (1) Products related to heat-killed preparations, such as GMNL-6 and GMNL-653 used in this study, can be a powerful remedy for healing injuries. (2) Heat-killed Lactobacilli offer excellent healing ability and the capability of preventing excessive fibrosis. (3) The suppression of TGF-β/pSmad signaling is involved in the skin wound repair promoting the function of heat-killed Lactobacilli. (4) The functional ingredient of heat-killed GMNL-6 or GMNL-653 preparation is referred to as LTA, one of the components of the cell wall.

## Figures and Tables

**Figure 1 cells-10-03264-f001:**
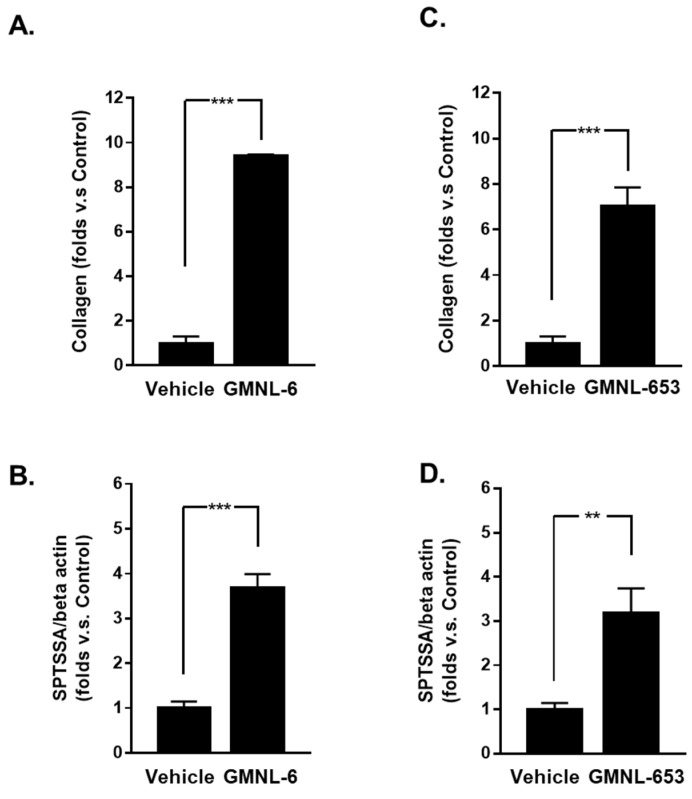
Heat-killed probiotic preparations of GMNL-6 or GMNL-653 promote collagen synthesis and increase the expression of SPTSSA in Hs68 cells. Hs68 fibroblast cells were seeded onto 6-well-plate at 2 × 10^5^ cells/well and treated with the heat-killed GMNL-6 (**A**,**B**) or GMNL-653 (**C**,**D**) as 109 cells/well for 24 h. (**A**,**C**) The synthesis of collagen protein in the culture supernatant was determined by an ELISA analysis. (**B**,**D**) The mRNA expression of SPTSSA was determined by SYBR Green-based quantified RT-PCR. All PCR data are normalized with β-actin, the internal control gene. (*n* = 3–5; values are mean ± SEM. ** *p* < 0.01, *** *p* < 0.001).

**Figure 2 cells-10-03264-f002:**
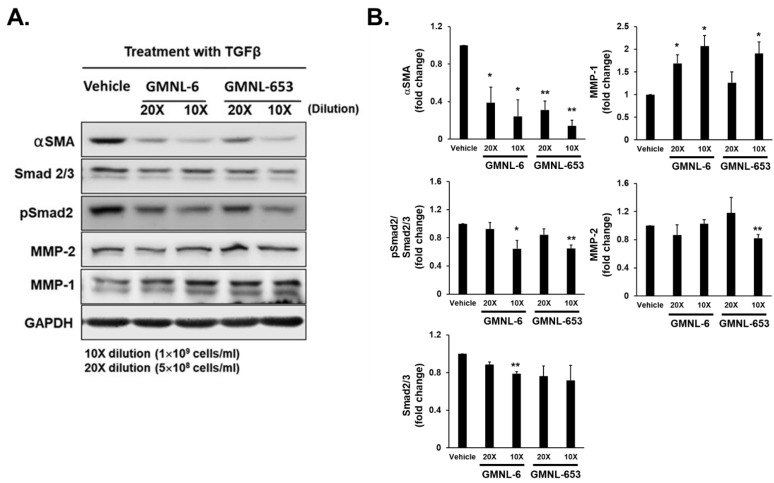
The heat-killed probiotics GMNL-6 or GMNL-653 preparation inhibits the TGF-β induced phosphorylation of Smad2 but increases the expression of MMP1 in Hs68 cells. Hs68 fibroblast cells were seeded onto 6-well-plates at a density of 2 × 10^5^ cells/well and treated with 20 ng/mL TGF-β in combination with the heat-killed GMNL-6 or GMNL-653 preparations at 10 fold (10×, 1 × 10^9^ cells/mL) or 20 fold (20×, 5 × 10^8^ cells/mL) dilution for 48 h. The proteins expressions of α-smooth muscle actin (αSMA), phosphorylated Smad2, total Smad2/3, MMP-1, or MMP-2 were determined by western blot analysis (**A**) and the bar graphs presented in (**B**) were resulted from two independent experiments. * *p* < 0.05; ** *p* < 0.01.

**Figure 3 cells-10-03264-f003:**
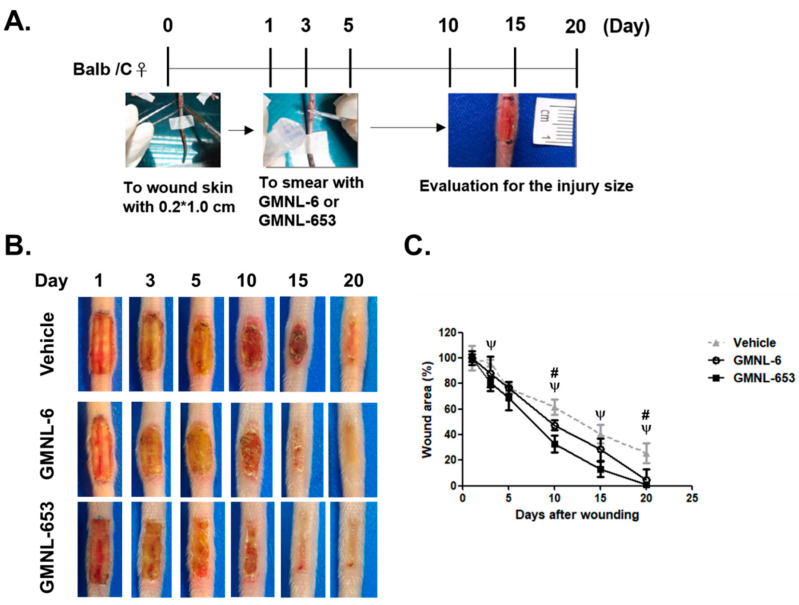
Heat-killed probiotics preparations of GMNL-6 or GMNL-653 exhibit excellent wound healing ability in mice. (**A**) The experimental scheme experimental skin wound of mice tails and treatment protocol using smeared gels with heat-killed GMNL-6 or GMNL-653 preparation. (**B**) The selected images of tail wounds were taken at the indicated day after wound introduction. Other images of each mouse used in this experiment were presented in Appendix A. (**C**) The quantified results of the tail wound images. For each group, *n* = 4; values present mean ± SEM; # or ψ presented as the *p* < 0.05 of GMNL-6 group or GMNL-653 group compared to the vehicle treatment group, respectively.

**Figure 4 cells-10-03264-f004:**
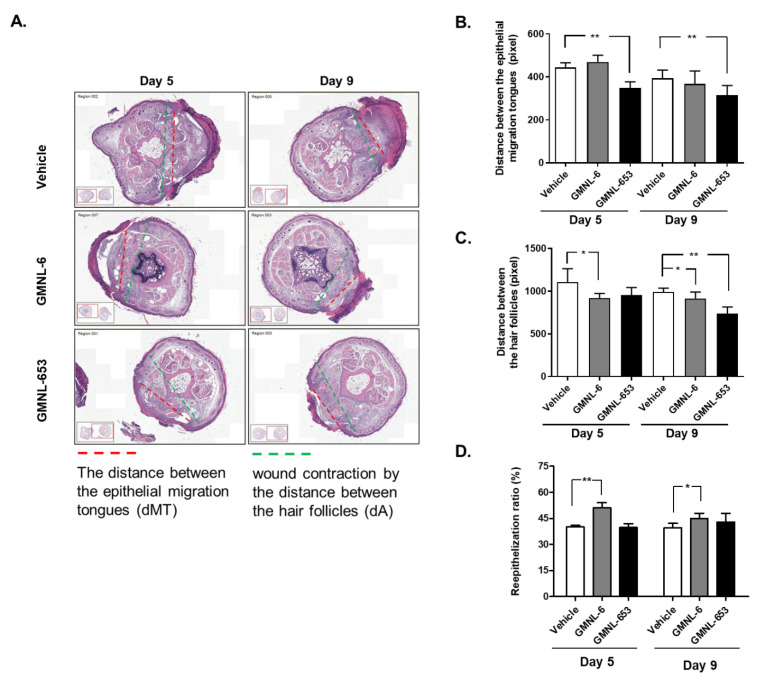
Probiotics GMNL-6 and GMNL-653 treatments provide prominent tail-wound healing recovery ability. The experimental tail wounds were introduced as the descriptions in Materials and Methods section. (**A**) The tail specimens were taped at Day 5 or Day 9 after wound introduction and performed H&E staining. The wound re-epithelialization or the wound contract was determined by the distance between the epithelial migration tongues (dMT, red dashed lines) or the distance between the hair follicles nearby the wound bed (dA, green dashed lines), respectively. (**B**,**C**) The quantitative results of dMT (**B**) or dA (**C**) were presented. (**D**) The reepithelialization ratio was calculated as the formula of dMT/dA and was presented as percentage of total wound length. For each group, *n* = 4; values present mean ± SEM, * *p* < 0.05, ** *p* < 0.01 when compared with the vehicle treatment.

**Figure 5 cells-10-03264-f005:**
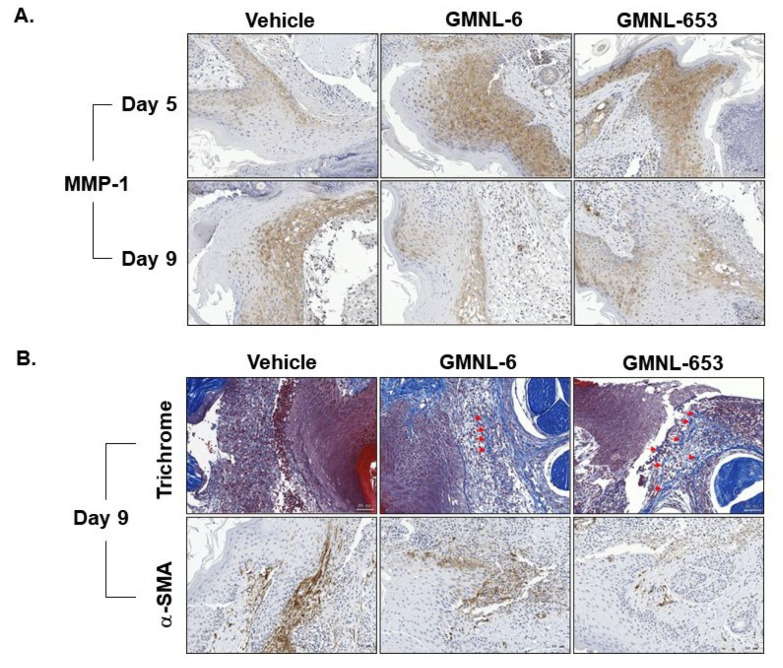
The treatment of heat-killed Lactobacillus GMNL-6 or GMNL-653 increases MMP-1 expression in the early stage and decreases αSMA expression at the later stage during the tail-wound healing. The experimental tail wounds were introduced as described in the Materials and Methods section and covered with gels that contained heat-killed GMNL-6 or GMNL-653 bacterial cells. The expressions of MMP-1 (**A**) or αSMA (lower panel in **B**) were determined by immunohistochemistry and the collagen fibers were determined by Trichrome stain (upper panel in **B**). Arrows in the upper panel of (**B**) indicated the structured collagen fibers.

**Figure 6 cells-10-03264-f006:**
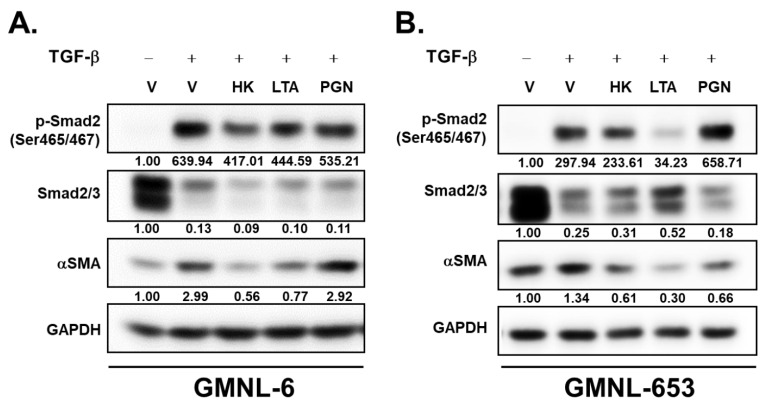
The treatment of lipoteichoic acid (LTA) caused similar benefits to those of heat-killed GMNL-6/GMNL-653 in TGF-β-induced fibrosis of Hs68 cells. The Hs68 cells were treated with 20 ng/mL TGF-β alone or in a combination of heat-killed (HK) Lactobacillus bacteria suspension at 1 × 10^9^ cells/mL, or the derived LTA at 50 μg/mL or peptidoglycan (PGN) at 50 μg/mL for 24 h followed by harvesting the total cellular proteins. The expressions of Smad2 phosphorylation and αSMA were determined by western blot. (**A**) GMNL-6, (**B**) GMNL-653. The inserted numbers indicated relative expression level in comparison to the vehicle (V) and without TGF-β treatment.

## Data Availability

Not applicable.

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
