# Peer review of "Heat-Killed Lactobacilli Preparations Promote Healing in the Experimental Cutaneous Wounds"

_cells, 2021, doi:10.3390/cells10113264_

Round 1

Reviewer 1 Report

Major comments:

1)  Why did the authors select Hs68 fibroblast cells?

2) What could be the explanation for lipoteichoic acid to achieve the anti-fibrogenic benefit similar to the heat-killed bacteria?

3) How (in which form) could the heat-killed format of Lactobacillus be used as an alternative approach to treating skin healing disorders?

4) Fig. 3A and B: images are too small

5) Fig. 4A: the distance was determined by the distance between the epithelial migration tongues (red 260 dashed lines) or the distance between the hair follicles (green dashed lines)

It is very difficult to understand how the distances were determined in the H/E images. Why no immunfluorescence was performed for example for hair follicles? Where ist the wound side?

Somewhat I do not believe in the distance differences presented in this Figure.

Author Response

  1. Why did the authors select Hs68 fibroblast cells?

Responses:

Fibroblasts play an important role in supporting wound repair, such as to secret collagen for the remodeling of extracellular matrix and to help wound contraction (Fibroblasts and wound healing: an update. Regen Med. 2018. 13(5):491-495.). For this reason, we selected Hs68 cells, the skin fibroblast cell line, as the cell model.

  1. What could be the explanation for lipoteichoic acid to achieve the anti-fibrogenic benefit similar to the heat-killed bacteria?

Responses:

The heat-killed preparation used in this study contained whole bacterial cells including cell wall. LTA is one of the major components of cell wall of gram positive bacteria like Lactobacilli. From Figure 6, the LTA could suppress the TGFb induced SMAD2 activation and aSMA expression, which is the possible anti-fibrogenic mechanism of LTA.

  1. How (in which form) could the heat-killed format of Lactobacillus be used as an alternative approach to treating skin healing disorders?

Responses:

We have added the rationale for this study in the introduction section as follows: “The topical use of probiotics in the improvement skin wound repair has been reported by several studies [22]. Most of the studies used live probiotics but it may become opportunistic infection when used in at-risk population groups, such as people with weaken immunity [23] and the heat-killed format could avoid such risk.” (page 2, line 69-72)

  1. Fig. 3A and B: images are too small

Responses:

We enlarge the Figure 3A. In order to make enlargement of Figure 3B, we put one picture of each selected time-point of each group and put others as a supporting data (Figure S1).

  1. Fig. 4A: the distance was determined by the distance between the epithelial migration tongues (red 260 dashed lines) or the distance between the hair follicles (green dashed lines)

Responses:

We have revised the descriptions in legends of Figure 4A as follows: “The wound reepithelization or the wound contract was determined by the distance between the epithelial migration tongues (dMT, red dashed lines) or the distance between the hair follicles nearby the wound bed (dA, green dashed lines), respectively.” (page 10, line 285-288). In addition, we revised the descriptions for the determinations of reepithelialization and wound contraction as follows: “We next evaluated the effect of heat-killed probiotics GMNL-6 or GMNL-653 gels to wound reepithelization or wound contraction by measuring the distance of migrating epithelial tongues (dMT) or the distance between hair follicles (dA), respectively, ac-cording to the report from Gerharz et al. [24] (Figure 4A).” (page 8, line 271-275).

  1. It is very difficult to understand how the distances were determined in the H/E images. Why no immunfluorescence was performed for example for hair follicles? Where ist the wound side? Somewhat I do not believe in the distance differences presented in this Figure.

Responses:

The measurement methods of dMT (the distance between the epithelial migration tongues) or dA (the distance between the hair follicles nearby the wound bed) is based on the descriptions of the article from reference No. 24 (Gerharz et al. Morphometric analysis of murine skin wound healing: standardization of experimental procedures and impact of an advanced multitissue array technique. Wound Repair Regen. 2007 Jan-Feb;15(1):105-12), which based on H&E staining to analyze. The quantitative data were statistically analyzed from four mice tail wounds as described in the Materials and Methods section. From the appearance of the wounds, one could find the improved effect of the gels containing heat-killed GMNL6 or GMNL-653 (Figure 3).

Reviewer 2 Report

This study reported the promoting effects of the heat-killed Lactobacilli probiotics on cutaneous wound healing. However, there are several concerns in this study for publication in Cells.

At the first, authors should validate the validity of models. They used a mouse tail wound model and provided the histology of wounds in Figure 4. However, their histology is remarkably different from that of the previous report such as Falanga et al. (WRR, 12: 320-326, 2004). Authors should discuss the excessive formation of granulation tissue and abundant abnormal tissue (scab?) on the surface of wounds. I am wondering in appropriate management of wound (dryness or infection) or chemical or osmotic irritation by gel. Authors should clearly mention what these animals are model.

  Authors provided the distance between the edge of epidermal tongues (dMT) to show the effects of probiotics on epithelialization. However, this indicator is affected by both epithelialization and wound contraction. Therefore, dMT should be corrected by the distance between wound edges (dA). The length of epidermal tongue is also direct indicator of epithelialization.

  Authors presented the decreased expression of alpha-SMA, which is an important factor for wound contraction, in probiotic treated animals by molecular biological and immunohistochemical methods. On the other hand, they provided enhanced wound contraction in the probiotic treated animals by the histological analysis. How do you explain this inconsistency?

  In the Introduction, authors should provide the hypothesis of this study and its rationale. What is the originality of this study? We can find a lot of similar articles, such as Nam et al. Nutrient. 13:2666, 2021, Tagliari et al. JPEN J Parenter Enteral Nutr. 2021, and so on.

  There are a lot of mistakes, for example, “aprobiotics” in keywords, “110% fetal bovine serum” on P3L107, and the Y-axis label of Fig. 4C.

Author Response

  1. At the first, authors should validate the validity of models. They used a mouse tail wound model and provided the histology of wounds in Figure 4. However, their histology is remarkably different from that of the previous report such as Falanga et al. (WRR, 12: 320-326, 2004). Authors should discuss the excessive formation of granulation tissue and abundant abnormal tissue (scab?) on the surface of wounds. I am wondering in appropriate management of wound (dryness or infection) or chemical or osmotic irritation by gel. Authors should clearly mention what these animals are model.

Responses:

In the article from Falanga et al., they used antibiotics after tail wound introduction. In our study, we did not apply antibiotics but using gel coverage to perform the basic wound management and that may be the reason about the observed large granulation tissues at Day 9 in vehicle group (Figure 4). Indeed, the rate of closure of experimental tail wounds in the vehicle group of our study was similar to the report from Crane et al. (J Vis Exp. 2020 Mar 25;(157):10.3791/60653.). In this revision, we added the detail descriptions about gel format and wound management in Materials and Methods section as follows: “We used 8-week old female BALB/c mice (n = 4 for each group) were anesthetized with isoflurane (3% for induction, 1% for maintenance) during surgery, and the wounds of 0.2 cm ´ 0.1 cm were introduced from 1 cm from the root of mice tails. The newly introduced wounds waited for one day to stop the aseptic gauze followed by covering gels (the gel used in this study was Ulora Gel (BEE BRAND MEDICO DENTAL CO., LTD., Osaka, Japan). The composition of Ulora Gel for 1 g was listed as followed: xanthangum 0.02 g, xylitol 0.06 g, trehalose 0.04 g, glycerol 0.1 g, sterilized H2O 0.28 g. with or without heat-killed GMNL-6 or GMNL-653 bacterial cells at a final concentration of 1´1010 cells/g of gel. For the experimental procedure, the tail wounds were smeared with 0.01g gel evenly per day (i.e. 1´108 bacterial cells/time) for a total of five days (from day-1 to day-5 after tail skin surgery). For make sure the completely absorbing of gels, it was stand for 15-20 minutes after spreading the gel. The tails of mice were imaged and recorded at 1, 3, 5, 10, 15, and 20 days after wounding.” (page 3, line 100-116)

  1. Authors provided the distance between the edge of epidermal tongues (dMT) to show the effects of probiotics on epithelialization. However, this indicator is affected by both epithelialization and wound contraction. Therefore, dMT should be corrected by the distance between wound edges (dA). The length of epidermal tongue is also direct indicator of epithelialization.

Responses:

As the report from reference No. 24 (Gerharz et al. Morphometric analysis of murine skin wound healing: standardization of experimental procedures and impact of an advanced multitissue array technique. Wound Repair Regen. 2007 Jan-Feb;15(1):105-12.), dMT is sufficient to describe the reepithelization. According to this reference, the definition of dMT is the distance between the epithelial migration tongues and the dA represents as the distance between the hair follicles nearby the wound bed. We have added the paper from Gerharz et al. as reference No. 24 and mentioned it in Materials and Methods section (page 4, line 162-166) and in the main-text (page 8, line 271-275).

  1. Authors presented the decreased expression of alpha-SMA, which is an important factor for wound contraction, in probiotic treated animals by molecular biological and immunohistochemical methods. On the other hand, they provided enhanced wound contraction in the probiotic treated animals by the histological analysis. How do you explain this inconsistency?

Responses:

The physiological function of myofibroblasts is to cause wound contraction thereby to reduce wound margin but these cells should be removed by apoptosis to prevent the excess wound contraction, scarring, and fibrosis (Clin Cosmet Investig Dermatol. 2014. 7:301-11.). However, the skin wound healing in Acta2 knockout mice was as good as the wild-type mice, which illustrates that myofibroblasts are unnecessary for wound contraction. (Lab Invest. 2015 Dec;95(12):1429-38.). We add some discussion about these observations as follows: “aSMA is a marker for myofibroblasts and the physiological function of these cells is to cause wound contraction thereby to reduce wound margin [32]. Myofibroblasts should be removed through the induction of apoptosis during the wound healing process to prevent the excess wound contraction, scarring, and fibrosis [32]. However, the skin wound healing in Acta2 knockout mice was as good as the wild-type mice, which illustrates that myofibroblasts are not absolutely necessary for wound contraction [33]. It is also known that there a heterogeneity of dermal fibroblasts that affects normal or pathological skin wound healing [34, PMID: 30062921]. Due to the observations of the reduction of aSMA expression in mouse tail wounds at Day-9 (Fig. 5) but accelerated the wound contraction (Fig. 4) by heat-killed GMNL-6 or GMNL-653, it suggests that heat-killed GMNL-6 or GMNL-653 may have different effects to different subpopulations of dermal fibroblasts but remains to be further investigated.” (page 13, line 371-382)

  1. In the Introduction, authors should provide the hypothesis of this study and its rationale. What is the originality of this study? We can find a lot of similar articles, such as Nam et al. Nutrient. 13:2666, 2021, Tagliari et al. JPEN J Parenter Enteral Nutr. 2021, and so on.

Responses:

The report form Nam et al (Nutrients. 13:2666, 2021) also topically used heat-killed probiotics preparation but the animal model used by the study was diabetic mice, and the underlying mechanism proposed by the authors was related to the metabolites from probiotics. However, we only took bacterial cells for preparing the heat-killed Lactobacillus solution indicating that there were no metabolites left. In our study, we further identify the functional composition of heat-killed Lactobacillus preparation in promoting skin wound repair, which is lipoteichoic acid. For the report from Tagliari et al., they used oral administration of probiotics, which totally differed from our model of topical application. To strengthen our manuscript, we have added the rationale for this study in the introduction section as follows: “The topical use of probiotics in the improvement skin wound repair has been reported by several studies [22]. Most of the studies used live probiotics but it may become opportunistic infection when used in at-risk population groups, such as people with weaken immunity [23] and the heat-killed format could avoid such risk.” (page 2, line 69-72)

  1. There are a lot of mistakes, for example, “aprobiotics” in keywords, “110% fetal bovine serum” on P3L107, and the Y-axis label of Fig. 4C.

Responses:

We thank the notices from the reviewer and apologize for such mistakes. We have carefully checked the accuracy of the revision form of manuscript.

Round 2

Reviewer 1 Report

The authors have succesfully improved the quality of their manuscript.

Author Response

We thank the reviewer to accept our manuscript for publication.

Reviewer 2 Report

In this revision, I don’t think the authors appropriately revised their manuscript for my major two concerns.

  1. Animal model

Basically said, the complete control group is required for the best experimental design. Based on the correct understanding the effects of vehicle from the comparison between the complete control and vehicle groups, we can find the effect of the target treatment. In this study, authors did not demonstrate the complete control, they must discuss the difference between the results of vehicle group and those of model animals previously reported (Falanga et al. WRR, 12: 320-326, 2004), and interpret the effects of vehicle treatment on the wound healing, at least.

  According to wound area reduction (Figure 3B), the wound area was increased from day 0 to 3 after wounding whereas Falanga et al reported gradual reduction of wound area from day 0 to 4 (Fig.3 in Falanga et al, 2004), which means some negative effect of vehicle gel on the inflammatory phase of wound healing. Their results showed both probiotics had protective effect from the negative impact of vehicle gel on the initial phase of wound healing.

From day 3 to 20, line graphs of the vehicle and GMNL-6 are almost parallel, whereas GMNL-653 showed more reduction rate, especially from day 5 to 10, compared with vehicle and GMNL-6. Authors can analyze such differences in the trends of wound area reduction with the interaction of treatment and time in a two-way ANOVA.

In addition, vehicle group showed the hyperplasia of granulation tissue and abundant necrotic tissue in Fig. 4A, whereas Falanga et al. (2004) did not reported such histological findings. Although the authors did not provide the high magnified images, abundant blue dots within the hyperplastic granulation tissue may indicated the remarkable infiltration of inflammatory cells. Authors must discuss the reasons why vehicle gel induced such abnormalities. I speculated that the vehicle gel has irritating effect on the tissue, and induced inflammation resulted in the initial deterioration in the inflammation phase and the hyperplasia of granulation tissue.

  Therefore, authors have to interpret that the difference in dA between vehicle and GMNL-6 might reflect the phase-gap due to the improvement of initial deterioration in vehicle group by GMNL-6. The results suggested that GMNL-6 has no-effect to promote wound healing except for the protective effect from vehicle gel in the initial phase. On the other hand, GMNL-653 may have a promoting effect on the proliferation phase of wound healing.

  1. Evaluation of epithelialization in histological analysis

dMT is an inappropriate indicator for epithelialization, because it becomes smaller when the wound contraction advanced even if epithelialization did not progressed. Authors must calculate the difference between dA and dMT, which indicate the total length of epidermal tongues.

Author Response

Responses to Reviewer#2

  1. Animal model

Basically said, the complete control group is required for the best experimental design. Based on the correct understanding the effects of vehicle from the comparison between the complete control and vehicle groups, we can find the effect of the target treatment. In this study, authors did not demonstrate the complete control, they must discuss the difference between the results of vehicle group and those of model animals previously reported (Falanga et al. WRR, 12: 320-326, 2004), and interpret the effects of vehicle treatment on the wound healing, at least.

  According to wound area reduction (Figure 3B), the wound area was increased from day 0 to 3 after wounding whereas Falanga et al reported gradual reduction of wound area from day 0 to 4 (Fig.3 in Falanga et al, 2004), which means some negative effect of vehicle gel on the inflammatory phase of wound healing. Their results showed both probiotics had protective effect from the negative impact of vehicle gel on the initial phase of wound healing.

From day 3 to 20, line graphs of the vehicle and GMNL-6 are almost parallel, whereas GMNL-653 showed more reduction rate, especially from day 5 to 10, compared with vehicle and GMNL-6. Authors can analyze such differences in the trends of wound area reduction with the interaction of treatment and time in a two-way ANOVA.

In addition, vehicle group showed the hyperplasia of granulation tissue and abundant necrotic tissue in Fig. 4A, whereas Falanga et al. (2004) did not reported such histological findings. Although the authors did not provide the high magnified images, abundant blue dots within the hyperplastic granulation tissue may indicated the remarkable infiltration of inflammatory cells. Authors must discuss the reasons why vehicle gel induced such abnormalities. I speculated that the vehicle gel has irritating effect on the tissue, and induced inflammation resulted in the initial deterioration in the inflammation phase and the hyperplasia of granulation tissue.

  Therefore, authors have to interpret that the difference in dA between vehicle and GMNL-6 might reflect the phase-gap due to the improvement of initial deterioration in vehicle group by GMNL-6. The results suggested that GMNL-6 has no-effect to promote wound healing except for the protective effect from vehicle gel in the initial phase. On the other hand, GMNL-653 may have a promoting effect on the proliferation phase of wound healing.

Responses:

We thank the comments from the reviewer. First of all, we would like to apologize the wrong labeling of Y-axis in the right panel of Figure 3B, which may lead the misunderstanding of the data. The correct labeling should be “Wound area (%)”, not “Wound area reduction (%)”. From the revised Figure 3B, the wound area of vehicle control on day3 remained at a similar level of that of day-1. Second, the difference between our data and the report of Falanga et al. (2004) may result from two reasons: (1) the use of antibiotics; (2) the use of mouse strain. For antibiotics issue, a systemic review by Altoé et al. (PLoS One. 2019. 14(10):e0223511.) suggested that the application of antibiotics in experimental wounds generally reduces wound healing time by the decreasing of inflammatory cells and increasing of fibroblast proliferation. We did not apply antibiotics in the tail wounds experiments, which may lead to the more infiltrated inflammatory cells in the vehicle control. The mouse strain used in the report of Falange et al. (2004) was 129SV´C57BL/6 but we used BALB/c. From the report of Li et al. (Heredity (Edinb). 2001 Jun;86(Pt 6):668-74), the wound healing rate of ear hole in 129J or C57BL/6 strain of mice was much higher than BALB/c. Similar observations were found in the healing of corneal wounds by the report from Pal-Ghosh et al., in which they found that the migration capability of corneal epithelial cells from BALB/c mice was 60% slower than those from C57BL/6 (Exp Eye Res. 2008. 87(5):478-86.).

We agree with the opinions from the reviewer in the difference between GMNL-6 and GMNL-653 on the promotion of skin wound healing. However, we think both strains of Lactobacilli display the beneficial effect on skin wound repair but the better one was GMNL-653, which was concluded from Figure 3.

To answer the comments from the reviewer, we perform some revisions as follows

  1. We re-run the statistical analysis of the wound area in Figure 3B by two-way ANOVA. (2.9 in Materials and Methods, line179)
  2. We make some discussions of the differences between ours and Falanga et al (2004) as follows: “Compared to the report from Falange et al. [32], the increased infiltrating inflam-matory cells and slower healing rate of tail wounds were found in the vehicle group of our study (Figure 3 and Figure 4). The differences may result from two factors. First, we did not apply antibiotics in wound management, but the study of Falange et al. used the antibiotics trimethoprim sulfa [32]. As a systemic review of the effects of an-tibiotics in animal skin wounds, Altoé et al. suggested that antibiotics generally re-duced wound healing time which was associated with reducing inflammatory cells and increasing the proliferation of fibroblasts [33]. Second, the mouse strains may cause a difference in wound healing rate. The mouse strain used in the study of Fa-lange et al. was 129SV´C57BL/6 [32], whereas we used BALB/c mice. From the report of Li et al., the wound healing rate of ear hole in 129J or C57BL/6 strain of mice was much higher than BALB/c [34]. Similar observations were found in the healing of corneal wounds by the report from Pal-Ghosh et al., in which they found that the migration capability of corneal epithelial cells from BALB/c mice was 60% slower than those from C57BL/6 [35]. Nevertheless, under the same wound management condition, heat-killed GMNL-6 or GMNL-653 gels showed good improvement.”(line341-356)
  • The better activity of heat-killed GMNL-653 than GMNL-6 was mentioned as follows: “However, GMNL-653 gel displayed a better activity in skin wound repair than GMNL-6. The significantly accelerated wound closure at day-3 and day-15 was only observed in the GMNL-653 group, but both GMNL-6 and GMNL-653 gels recovered the tail wounds at day-20, whereas the wound area remained more than 20% in the ve-hicle group (Figure 3C).” (line 246-250)
  1. Evaluation of epithelialization in histological analysis

dMT is an inappropriate indicator for epithelialization, because it becomes smaller when the wound contraction advanced even if epithelialization did not progressed. Authors must calculate the difference between dA and dMT, which indicate the total length of epidermal tongues.

Responses:

We agree with the comments from the reviewer. The ratio of reepithelialization was calculated by dMT/dA (i.e. the distance between epithelial migration tongues divided by the wound width) as the suggestion from the reviewer in revised Figure 4D. From Figure 4D, the increased reepithelialization ratio was observed in GMNL-6 group at day-5 and day-9, but it was stayed at similar levels in GMNL-653 group in comparison to vehicle control.

We revise the descriptions of histological analysis of Figure 4 as follows: “We next evaluated the effect of heat-killed probiotics GMNL-6 or GMNL-653 gels to the distance of migrating epithelial tongues (dMT) or the distance between hair folli-cles (dA) respectively, according to the report from Gerharz et al. [24] (Figure 4A to 4C). The reepithelialization ratio of wounds was calculated by dMT/dA (Figure 4D). The quantification results revealed that gels containing the heat-killed probiotics GMNL-6 or GMNL-653 could effectively reduce the dMT measurement from day-5 to day-9 (Figure 4B). Similar results were observed for wound contraction, which was evaluated by dA measurements (Figure 4C). However, the ratio of reepithelialization was increased in the GMNL-6 group at day-5 and day-9 (Figure 4

Round 3

Reviewer 2 Report

I think authors appropriately responded to my comments, but they forgot to correct the legend of figure 4.